# Liquid-Modulated Photothermal Phenomena in Porous Silicon Nanostructures Studied by μ-Raman Spectroscopy

**DOI:** 10.3390/nano13020310

**Published:** 2023-01-11

**Authors:** Oksana Makukha, Ivan Lysenko, Ali Belarouci

**Affiliations:** 1Lyon Institute of Nanotechnology, UMR 5270, INSA de Lyon, 69100 Villeurbanne, France; 2Physics Department, Taras Shevchenko National University of Kyiv, 01033 Kyiv, Ukraine

**Keywords:** thermal conductivity, porous silicon-liquid composite, μ-Raman spectroscopy, photothermal phenomena, liquid-confined nanopores, silicon nanostructures

## Abstract

In the present study, the effect of liquid filling of the nanopore network on thermal transport in porous Si layers was investigated by μ-Raman spectroscopy. The values of thermal conductivity of porous Si and porous Si-hexadecane composites were estimated by fitting the experimentally measured photoinduced temperature rise with finite element method simulations. As a result, filling the pores with hexadecane led to (i) an increase in the thermal conductivity of the porous Si-hexadecane composite in a wide range of porosity levels (40–80%) and (ii) a suppression of the characteristic laser-induced phase transition of Si from cubic to hexagonal form.

## 1. Introduction

The interest in engineering thermal transport properties is consistently emerging for a wide range of micro- and nanodevices and materials-based energy technologies. This demand is predominantly being pushed by two specific technological challenges. Firstly, a high thermal conductivity is required for cooling nanoscale components or for efficient interfacial thermal transport, where the heat can dissipate fast and efficiently and, accordingly, their high-operation speed, power, stability and reliability can be preserved. On the other hand, a low thermal conductivity is targeted for thermoelectric energy conversion to achieve a high figure of merit (ZT).

With the progress of nanotechnology and the ability to fabricate structures down to a few nanometers in size, the past two decades have seen a great interest in understanding fundamental properties of materials with structural characteristic lengths on the nanometer scale, which are close to, or even smaller than, the intrinsic length scales (such as mean free path and wavelength) of elemental energy carriers (such as phonons, electrons and photons). Despite the fact that cutting-edge approaches for manipulating electronic and photonic transport have been proposed in the past years, progress on controlling lattice vibrations (i.e., the phonons) is yet significantly behind. However, in the past two decades, rapid developments in synthesis and processing of nanoscale materials have created a great demand for understanding thermal transport in low-dimensional nanomaterials [1,2,3,4,5,6,7]. Nanostructures, including one-dimensional (1D) structures, like nanotubes (NTs) [8] and nanowires (NWs) [9], two-dimensional (2D) materials, like graphene [10], and thin films consisting of alternating layers, superlattices [11,12], have been investigated.

Silicon nanostructures with a high density of nanoscale features, such as interfaces, porosity and impurities have thermal conductivities (κ) up to three orders of magnitude lower than bulk silicon through enhanced phonon scattering [13,14,15,16,17]. In particular, by adjusting the porosity and the pore size of porous silicon, the thermal conductivity approaches the amorphous limit (0.2 to 0.5 Wm^−1^K^−1^) [18,19]. Porous silicon was originally studied for optical purposes, but the large internal surface area (up to 900 m^2^/cm^3^), the adjustable porosity (up to 90%) and pore sizes (from 1 nm to 10 µm), and a tunable surface chemistry, make it a promising nanomaterial for biology, chemistry and medicine [20,21].

However, in recent years, a considerable amount of interest has been dedicated to the investigation of the thermal properties of porous silicon in air; the effect of confined liquid on the thermal transport properties of the system has not been extensively explored. Not only fundamental theories, but also many important experimental observations, are still lacking. A comprehensive understanding of the heat conduction in liquid-confined porous silicon is important for a variety of applications, such as desalination devices, flow sensing and nanofluidics. The main mechanisms involved in the formation of photoacoustic signals in porous silicon-liquid composites, and constrictions caused by filling of the pore network have been reported in [22,23]. The increase of the thermal conductivity (up to two times) of the composite system “porous silicon—viscous liquid”, in comparison with pristine porous silicon, has been experimentally confirmed by means of the photoacoustic technique [24]. Several theoretical analysis of thermal transport in porous Si and porous silicon-water nanocomposite with quantitative estimation of thermal conductivity values were performed using a molecular dynamics approach [25,26,27].

In this paper, we present an experimental study of the impact of an oil-like liquid on several photothermal effects, including: (i) photo-induced heat transport in porous Si-liquid nanocomposite and (ii) photo-induced phase transition in porous Si layers by means of micro-Raman spectroscopy.

## 2. Materials and Methods

### 2.1. Sample Preparation

Porous silicon layers were prepared by electrochemical anodization of p-type boron doped (100)-oriented silicon wafers (resistivity ∼10–20 mΩ·cm). Silicon wafers were cut into 1.5 × 1.5 cm pieces, cleaned with acetone and ethanol in an ultrasonic bath, and then blow-dried with nitrogen. The etching process was performed at room temperature, at current densities in the range of 40–250 mA·cm^−2^ in a HF (48%)—ethanol solution (1/1, *v*/*v*). An etch stop technique [28,29] was applied to ensure the best in-depth homogeneity of the porous layers. Permanent stirring was applied to remove hydrogen bubbles that usually appear during the anodization process. At the end of the etching process, samples were rinsed several times with ethanol and dried under nitrogen flow. Under these conditions, meso-porous silicon layers with pore diameters ranging from 2 to 50 nm, and porosity-dependent specific surface areas varying from 120 to 200 m^2^/g, were formed. The thickness of the formed porous layers was approximately 100µm, measured by Scanning Electron Microscopy (SEM). The porosity of the samples being in the range 40–80% was estimated by means of infrared reflectivity [30]. N-Hexadecane (99%, Chimie Plus Laboratories, Saint Paul de Varax, France) was chosen as a liquid medium because of its favorable physical/chemical properties, ensuring even and complete filling of the pores: low viscosity (~3 mPa·s), no significant evaporation on the timescale of the measurements (boiling temperature 286.9 °C, saturation vapor pressure ~0.3 Pa) and chemical inertness toward silicon surfaces. Thermal conductivity 0.147 Wm^−1^K^−1^ is close to the values of significant parts of other liquid substances in the considered temperatures range. The filling factor of pores with hexadecane was controlled by infrared reflectivity as described below.

### 2.2. Estimation of Porosity and Filling Factor of Nanopores

The porosity and the degree of pores’ filling (hexadecane filling factor) were estimated using the Looyenda-Landau-Lifshotz (3L) model [30]. Reflectivity spectra have been recorded in the 1400–1700 nm wavelength range with a near-infrared NIRQuest spectrometer (Ocean Optics, Ostfildern, Germany), before and after filling the pores with hexadecane. The thickness of the porous silicon layers was preliminarily estimated from SEM images (see Appendix A).

### 2.3. Micro-Raman Spectroscopy

Raman spectra were recorded with a Jobin-Yvon Aramis spectrometer. The incident light was emitted by a diode laser at 473 nm. The laser beam was focused down to a spot diameter of 1.5 µm on the surface of porous Si samples with a 50× magnification and 0.45 numerical aperture microscope objective. All reported intensities, positions and linewidths of the measured Raman peaks have been extracted by Lorentzian fittings.

#### 2.3.1. Estimation of Nanocrystal Size Distribution

The size distribution of nanocrystals forming a porous Si layer was estimated from fitting the spectral signature of the cubic Si LO phonon mode in the Raman signal obtained at low power values of the exciting laser. A phenomenological model already reported in detail earlier [31,32] was used to deduce a Gaussian distribution of the nanocrystallite sizes.

#### 2.3.2. Local Temperature Measurements

The absorbed laser light focused on the surface of a porous Si layer works as a heating power source and generates a temperature gradient through the whole structure, depending on its thermal conductivity and on the absorption depth (1–1.5 µm for the case of porous Si with porosities in the range: 40–80%). The resulting laser-induced temperature rise at the layer surface leads to a shift of the Raman peak towards lower wave numbers [33,34]. A preliminary relationship between the Raman peak position and sample temperature was determined at relatively low laser powers to avoid any additional laser-induced heating. The obtained linear calibration function with a slope of −0.025 cm^−1^/K was used to estimate the temperature rise values, as shown in Appendix A.

### 2.4. Finite-Element Heat Modeling

Finite-element 3D simulations of photo-induced heat transport in porous Si layers were performed with the commercial Ansys Lumerical Heat module. Typical tetrahedral mesh sizes used for the 3D heat transport simulations were chosen to be in the 0.1–20 µm range for the corresponding thickness of porous layers in the range of 70–100 µm. The steady state Fourier heat conduction equation was numerically solved. The resulting spatial distribution of the temperature induced by laser heating at the surface of porous Si layers was calculated in correlation with the Raman measurements described above, with a thermal conductivity value of the porous Si layer used as a single-fitting parameter.

## 3. Results

### 3.1. Nanostructural Properties of the Porous Si Layers

A typical view of an anisotropic morphology of a meso-porous Si nanostructure formed by electrochemical etching of a highly doped p-type Si wafer is shown in Figure 1a) As one can see, the porous layer consists of a dendritic columnar-like silicon skeleton, with an average diameter of nanocrystallites of the order of 10 nm. The nano-columns are perpendicular to the wafer surface and separated by empty pores. These quasi-columns remain mono-crystalline and retain the crystalline orientation of the original Si substrate. In order to estimate more precisely the size distribution of the nanocrystallites forming the porous layer, the corresponding Raman spectrum obtained at low laser power (<0.1 mW) was fitted according to the model reported earlier [31,32].

Figure 1b shows an example of the theoretical curve (red line) fitting the experimental Raman spectrum (open circle symbols). As a result of the fitting, a Gaussian distribution of the Si nanocrystallite sizes can be obtained (see inset in Figure 1b). Table 1 summarizes the main structural data of the porous Si samples used in this work.

### 3.2. Impact of Hexadecane on Photo-Induced Heat Transfer in Porous Si

As one can see in Figure 2a, the main Raman peak (corresponding to the cubic Si LO phonon mode) of the typical porous Si layers with empty pores shifts towards lower phonon frequencies when the absorbed laser power increases. The laser heating-induced red-shift of the Raman peak as high as 10 cm^−1^ over the 0.02–0.9 mW range of the absorbed laser powers is also accompanied by a characteristic spectral broadening of the peaks. 

Using the calibration curve displayed in Appendix A, one can convert the laser-induced spectral position of the Raman peaks to the corresponding temperature rise values. Figure 2b illustrates the temperature evolution under the laser spot as a function of the absorbed laser power for the porous layer with empty pores as well as with the pores filled with hexadecane. In both cases, linear dependencies were obtained. The lower slope characterizing the porous sample containing hexadecane in its pores can be explained by a higher thermal conductivity value of the porous Si/hexadecane composite.

In order to estimate thermal conductivities of porous Si layers and porous Si/hexadecane composites, laser-induced thermal gradients in the porous Si without and with hexadecane were simulated and illustrated in Figure 3a,b, respectively.

The local values of temperature rise were estimated initially from micro-Raman spectroscopy (see Figure 2b) and achieved inside a near-surface cylindrical region of the laser spot with 1.5 µm diameter. The single-fitting parameter, allowing a precise correlation between the experimentally measured and theoretically simulated temperature rise values, was the thermal conductivity of the porous Si layers without and with hexadecane. For the case of thermally thick, porous Si layers, shown in Figure 3, the layer thickness (70–100 µm) being much larger than the laser beam diameter (1.5 µm) results in the almost-perfect semispherical temperature isotherms for porous Si both without and with hexadecane. This means that the laser beam focused on the porous Si surface can be considered as a point heating source. 

Figure 4 shows the porosity dependent evolution of the thermal conductivity of porous Si layers with empty pores (being in good agreement with the experimental data published earlier [18,24,33,34,35]), and pores filled with hexadecane.

As one can see, the thermal conductivity values monotonously decrease for both cases. At the same time, the thermal conductivity values characterizing the porous Si/hexadecane composites (k_pSi−hex_) are systematically higher, compared to the values for the porous Si layers with empty pores (k_pSi−air_).

This fact can be explained by the higher thermal conductivity value of hexadecane (0.147 Wm^−1^K^−1^) localized in pores in comparison with air (0.024 Wm^−1^K^−1^). As for the observed porosity dependent decrease of thermal conductivities, both for porous Si with empty pores and for the porous Si/hexadecane composites, it emphasizes preferential heat transport through the silicon skeleton for both cases. As one can see from the data presented in Table 1 and SEM images (see Figure 1a and Appendix A), the characteristic size of silicon nanocrystallites decreases with the increasing of porosity level. As it has been demonstrated and reported earlier in [23,36], thermal transport through a porous silicon layer was found to be strongly affected by the fine solid-state Si constrictions, while the presence of liquid in the nanopores results in the appearance of a new heat transfer channel (through the filled pores) between the Si crystallites [23]. Indeed, filling the pores of 42.5% of porous samples with hexadecane does not lead to a significant increase of the thermal conductivity of the composite: k_psi−air_ = 3.1 Wm^−1^K^−1^ and k_psi−hex_ = 3.4 Wm^−1^K^−1^ for air in pores and hexadecane-filled sample, respectively (ratio k_psi−hex_/k_psi−air_ = 1.1). At the same time, for the sample with the highest porosity (78.7%), the ratio between k_psi−hex_/k_psi−air_ reaches the level of 1.7. However, the continuously increasing porosity dependent ratio k_pSi−hex_/k_pSi−air_ indicates a progressive increase of the hexadecane role in the heat transport of the nanocomposites, due to: (i) significant reduction of the heat transport via interconnected Si nanocrystallites constituting the porous layer and (ii) increase of the global pore volume filled with hexadecane for the porous layers with higher porosities. Despite only hexadecane being used in this work, obtained results reported in this paper can be generalized to almost any liquid with similar thermal conductivity.

### 3.3. Impact of Hexadecane on Photothermally Induced Polytype Transition in Porous Si

Figure 5a shows one-phonon Raman spectra of a 78.7%-porous Si layer with empty pores, and pores filled with hexadecane at room temperature and under low-level photoexcitation (at a very low 0.03 mW absorbed power). Both spectra are completely identical and display a single asymmetric peak centered near 518 cm^−1^, which corresponds to the well-known Raman peak of cubic Si, with slightly increased lattice constant. Obviously, the presence of hexadecane has no impact on the crystalline nature of the Si skeleton forming the porous layer.

On the contrary, the Raman spectra at a much higher excitation level (2.15 mW), shown in Figure 5b, are significantly different. They can be easily deconvoluted into several Lorentzian-like peaks: (i) heated cubic phase (at 512 cm^−1^); (ii) hexagonal phase (495 cm^−1^); and (iii) amorphous phase (very large band centered at 475 cm^−1^) [37]. The hexagonal phase dominates in the porous Si samples with empty pores, while the Raman spectrum of the porous Si/hexadecane composites is mainly defined by the cubic phase spectral band.

A similar photo-induced phase transition effect was already observed on other types of Si nanostructures, as reported earlier in the literature [35,37,38]. The photo-induced formation of the hex-Si phase is known to be stimulated by strong mechanical stresses, appearing in a homogeneously heated nanostructured Si-based sample. A relatively strong local laser-induced heating of Si nanostructures plays a key role in the phase transition phenomenon. Since this condition can be easily achieved on a thermally thick, porous Si layer, one can observe the effect illustrated by the domination of the hexagonal phase in the corresponding Raman spectrum. The presence of a liquid in the nanopore network (hexadecane, in our case), leading to an enhancement of the effective thermal conductivity of the composite, induces a considerable decrease of the laser-induced temperature rise. The temperature at which the formation of hexagonal phase occurs is estimated to be more than 400 K [38]. Thus, the required temperature threshold cannot be reached, and the dominant crystalline phase will remain cubic.

## 4. Conclusions

Several photothermal effects in porous Si layers and porous Si-hexadecane nanocomposites were studied and compared. First, the presence of hexadecane in the nanopore network of a porous Si layer significantly impacts its photo-induced heat conduction properties, as was shown by means of micro-Raman spectroscopy coupled with finite element simulations. In particular, thermal conductivity values characterizing the porous Si/hexadecane nanocomposites were found to be systematically higher, compared to the values for the porous Si layers with empty pores in the porosity range of 40–80%. The porosity dependent enhancement of the hexadecane impact on the heat transport in the nanocomposites was experimentally demonstrated. Second, the filling of the nanopores with hexadecane leads to the suppression of the pronounced laser-induced phase transition of Si from cubic to hexagonal, as previously observed in various types of Si nanostructures.

## Figures and Tables

**Figure 1 nanomaterials-13-00310-f001:**
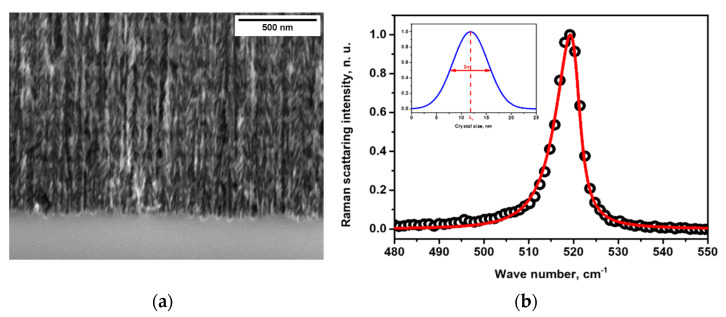
(**a**) SEM picture of a typical porous Si layer with 70% porosity; (**b**) fitting (red line) of a typical Raman spectrum (open circles) of a porous Si layer, in terms of a model allowing the estimation of the Gaussian-shape size distribution (blue curve) of Si nanocrystallites forming the porous layer.

**Figure 2 nanomaterials-13-00310-f002:**
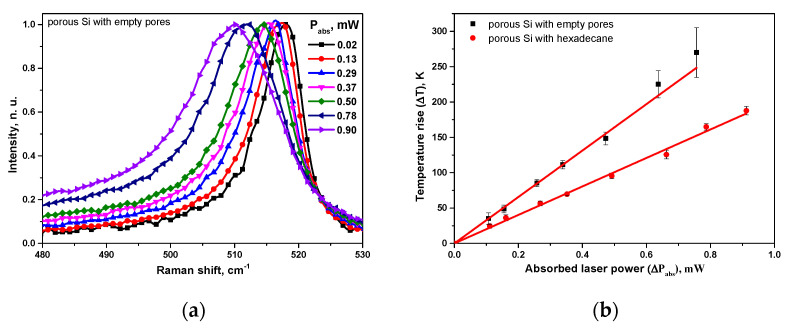
(**a**) Raman peak position of a porous Si layer of 80% porosity with empty pores at various absorbed laser power; (**b**) temperature rise induced by absorbed laser power for porous Si layers with empty pores and pores filled with hexadecane (4 repetitions were performed for each experimental point).

**Figure 3 nanomaterials-13-00310-f003:**
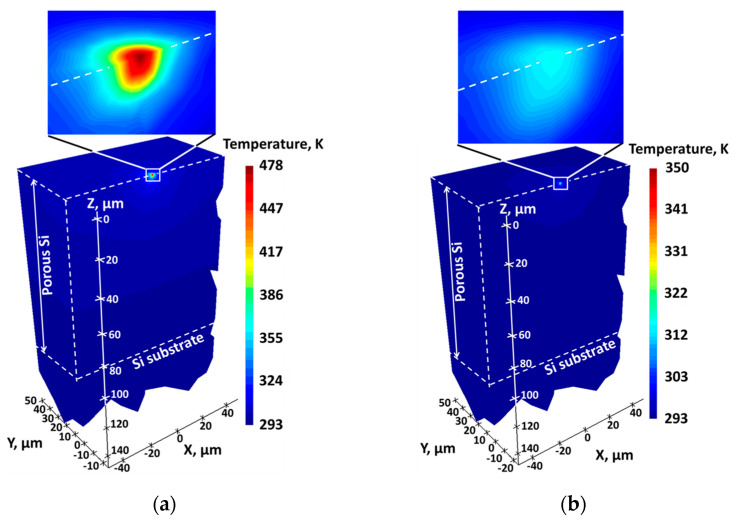
Simulated laser-induced temperature distribution in: (**a**) porous Si layers with empty pores; and (**b**) porous Si/hexadecane composites.

**Figure 4 nanomaterials-13-00310-f004:**
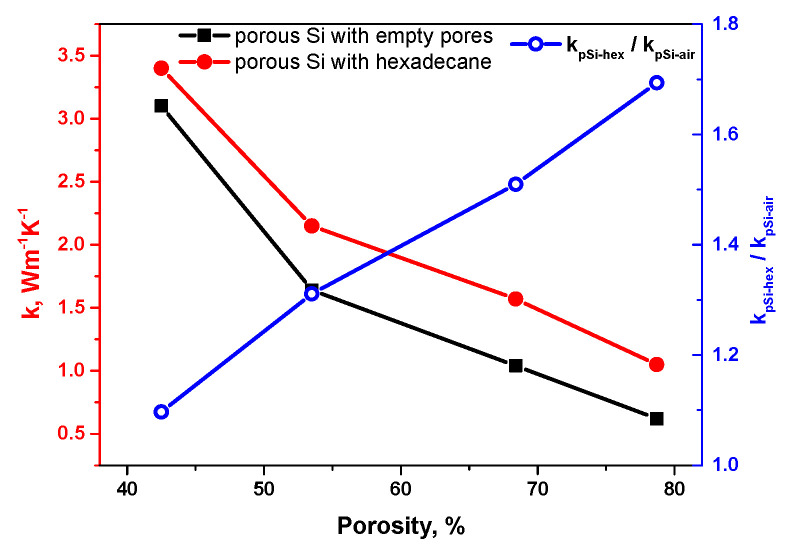
Porosity dependent thermal conductivity of porous Si layers without and with hexadecane in pores.

**Figure 5 nanomaterials-13-00310-f005:**
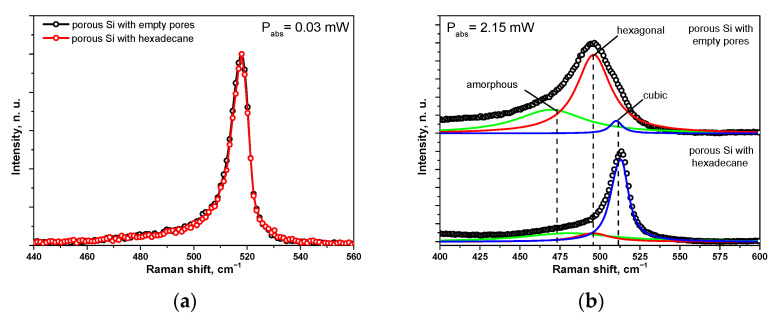
Raman peaks of a porous Si layer of about 80% porosity with empty pores, and pores filled with hexadecane at low and high absorbed laser powers: (**a**) 0.03 mW and (**b**) 2.15 mW.

**Table 1 nanomaterials-13-00310-t001:** Porosity-dependent size distribution of Si nanocrystallites.

Porosity, %	Thickness, μm	Size of Si Nanocrystallites, nm
42.5	71	11.8 ± 3.5
53.5	94	9.6 ± 3.0
68.4	115	9.3 ± 2.7
78.7	111	8.3 ± 2.7

## Data Availability

Not applicable.

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
