# Peer review of "Liquid-Modulated Photothermal Phenomena in Porous Silicon Nanostructures Studied by μ-Raman Spectroscopy"

_nanomaterials, 2023, doi:10.3390/nano13020310_

Round 1

Reviewer 1 Report

This paper investigated the thermal transport in porous Si layer by Raman Spectroscopy. It is overall interesting and can be published provided that some major revision being made.

1. For the title, "Liquid-modulated" may be not so accurate. And there is no more discussion on the influence of different liquid except hexadecane.

2. For the finite-element simulation, what kind of pores are used and what is the boundary condition? Also, what is the thermal conductivity of silicon and hexadecane? In Fig. 3, could the pores be clearly indicated?

3. For Fig. 5b, the meaning of different curves are not so clear. And it is confusing that the authors claimed that an enhancement of the effective thermal conductivity of the composite would induce a considerable decrease of the temperature under the laser spot, the required temperature threshold cannot be reached, and the dominant crystalline phase will remain cubic. Did the authors made some detailed investigations?

4. Most importantly, filling the pores with a high thermal conductivity hexadecane would obviously increase the thermal conductivity of porous silicon. The novelty of the paper is not so clear. 

Reviewer 2 Report

In this manuscript, the authors present an experimental study of the impact of an oil-like liquid on several photothermal effects including: (i) photo-induced heat transport in porous Si-liquid nanocomposite and (ii) photo-induced phase transition in porous Si layers by means of micro-Raman spectroscopy. In my opinion, this manuscript is interesting to the readers of Nanomaterials. The topic is very important in this field. This work is novel and original. The authors have solid background in this field. Therefore, the referee recommends it to be published after the following revisions:
1. The English should be polished by a native speaker.
2.
It is suggested the authors to check their manuscript carefully and thoroughly to avoid too many typical mistakes and mistypes.

3. The referee suggests enriching the discussion based on the experimental results, which will be very important for the readers in the relative field.

4. In Figure 1(a), the authors should provide different SEM pictures with different% porosity.

5. In Figure 2(b), how many data points are used for drawing the error bar?

In general, this work seems to be very interesting. The referee would like to see the revision if possible.

Reviewer 3 Report

The authors of the article studied several photothermal effects in porous Si layers and porous Si-hexadecane nanocomposites using micro-Raman spectroscopy in combination with finite element modeling. It has been shown that the presence of hexadecane in the nanopore  network of  a porous Si layer significantly affects its photo-induced heat conduction  properties. The porosity dependent enhancement of hexadecane's effect on heat transfer in nanocomposites has also been experimentally demonstrated.  It was also shown that filling nanopores with hexadecane leads to suppression of a specific laser-induced Si phase transition from cubic to hexagonal, which was previously observed in various types of Si nanostructures.

The work undoubtedly deserves to be published in the journal Nanomaterials.

Author Response

Dear reviewer,

thank you for your work and comments. We appreciate it a lot and all your remarks were taken into account.

Round 2

Reviewer 1 Report

The authors have corresponded to all my concerns and I recommend the acceptance of the paper.